# Does continuing learning help middle-aged and elderly people find employment in China? A propensity score matching analysis

Yuan Tao[1], Kexin Ren [2]*

1 College of Mathematics and Computer, Jilin Normal University, Siping, Jilin, China, 2 College of Physical Education, Jilin Normal University, Siping, Jilin, China

* jlsdrkx@163.com

## Abstract

### Background

The global aging workforce, driven by declining fertility and increased longevity, poses challenges to socio-economic systems and highlights the need for policies addressing age-related employment barriers. This also emphasizes the potential role of continuing learning in enhancing job prospects for older adults.

### Methods

Our data were derived from waves 3–5 of the China Health and Retirement Longitudinal Survey (CHARLS), encompassing 43,357 residents aged 45 years and older. The study defines employment based on responses to work-related questions and assesses the impact of continuing education or training courses. We employed propensity score matching (PSM), controlling for 10 confounding factors, to analyze the effects of continued learning on the employment of middle-aged and older individuals.

### Results

Of the 43,357 samples, 68.2 percent of Chinese individuals over 45 years of age were employed, but only 1 percent of this population engaged in continuing education. Initially, whether or not to pursue further studies did not seem to affect employment among middle-aged and older individuals. However, after eliminating endogenous selection bias through PSM, we found that participation in continuing education positively impacts the employment of middle-aged and elderly people. This result was verified through multiple matching methods.

### Conclusion

The findings underscore the importance of continuing learning in facilitating the employment prospects of middle-aged and elderly individuals.

**Data availability statement:** All data files are available from the ICPSR database (https://doi.org/10.3886/E223201V1)

**Funding:** This study was funded and organized by the Research Projects on Teaching Reform of Vocational and Adult Education in Jilin Province (2023ZCY302), the Scientific Research Program of Jilin Provincial Department of Education (JJKH20240541SK), and the Scientific Research Program of Jilin Provincial Department of Education (JJKH20250916SK).

**Competing interests:** The authors have declared that no competing interests exist.

## Introduction

The global population is experiencing a demographic shift characterized by a growing proportion of elderly individuals. This trend is primarily driven by declining fertility rates and increasing life expectancy [1,2]. As a result, the workforce in many countries is aging, presenting various socio-economic implications. For instance, there is a potential strain on pension systems [3] and healthcare services [4,5] due to the increased demand from elderly individuals. Additionally, there may be challenges in sustaining economic growth and productivity as the labor force ages, leading to concerns about future workforce shortages and reduced innovation [6]. These socio-economic implications highlight the need for proactive policies and strategies to address the aging workforce phenomenon.

Middle-aged and elderly individuals often face barriers to employment despite their valuable skills and experience. Technological advancements and changing job requirements have led to a mismatch between the skills possessed by older workers and those demanded by the labor market [7]. Moreover, age discrimination remains prevalent in many workplaces, with employers showing bias against older job seekers [8,9]. This discrimination can manifest in various forms, such as reluctance to hire or promote older workers, stereotyping based on age, and unequal access to training and development opportunities. As a result, middle-aged and elderly individuals may experience difficulties in finding or retaining suitable employment, leading to economic insecurity and social exclusion [10].

Therefore, how to improve the employment opportunities of middle-aged and elderly people is an important issue. Does continuing learning help middle-aged and elderly people find employment? This question is not only of academic interest but also carries practical implications for labor market policies and workforce development strategies. Understanding the relationship between continuing learning initiatives and employment outcomes for this demographic cohort is crucial for devising effective interventions to promote workforce participation and economic inclusion among older adults.

This study based on the China Health and Retirement Longitudinal Study (CHARLS), aimed to explore the impact of continuing learning on employment outcomes among middle-aged and elderly individuals using a propensity score matching (PSM) analysis. Specifically, we focus on the following research questions: (1) Does participation in continuing learning programs have a significant impact on the employment outcomes of middle-aged and elderly individuals? (2) If so, what are the potential mechanisms through which continuing learning affects employment outcomes? PSM allows us to compare the employment outcomes of individuals who participate in continuing learning programs with those who do not, while accounting for potential selection bias. By employing rigorous statistical methods, we aim to provide empirical evidence on the effectiveness of continuing learning initiatives in facilitating employment for middle-aged and elderly populations.

## Methods

### Data

The data in this study come from CHARLS, a nationally representative longitudinal survey of Chinese people. The study was approved by the Ethics Committee of Peking University (IRB00001052–11015). The CHARLS national baseline survey was conducted in 2011, covering approximately 17,000 individuals in 150 counties, 450 villages, and 10,000 households. Respondents were aged 45 years or older, and their spouses were included. CHARLS used a multistage, stratified, probability proportional to size (PPS) sampling methodology, with follow-up visits every two years from 2013 to 2020. The response rates of all samples were higher than 80% in each wave [11]. This study utilised CHARLS data from 2015, 2018 and 2020. As needed, we cleaned the data as follows.

Firstly, we matched and merged data from some modules in CHARLS. Secondly, we deleted observations under 45 years of age and retained those with data from three time periods, and finally deleted missing values, the final sample size left for analysis was 43,357. (See Fig 1)

### Variable measurement

**Dependent variable.** The CHARLS database asks the following questions about work: "Did you engage in agricultural work for at least 10 days in the past year for your own household?", "Did you work for other farmers/employers and get paid for at least ten days in the past year?", "Not including agricultural work, did you work for at least one hour last week in paid work, individual business or family business without getting paid?". Those who answered "yes" to any of the three questions above are considered to be employed and are assigned a value of "1", otherwise they are assigned a value of "0".

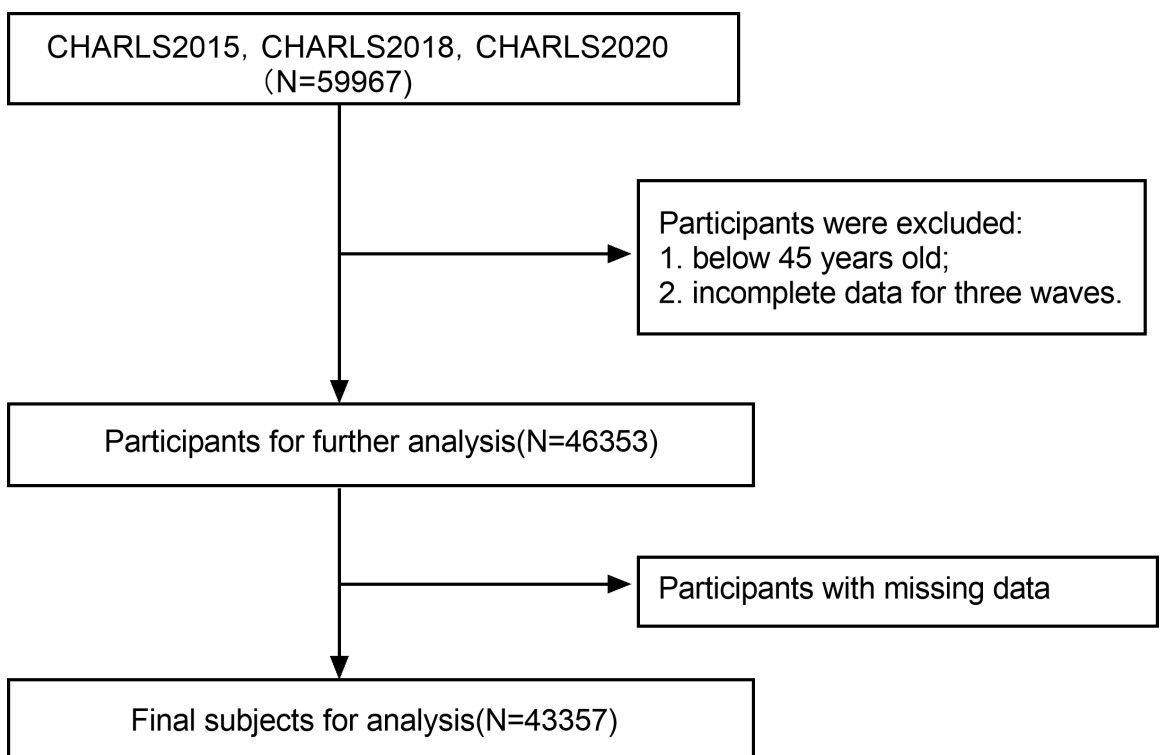

**Fig 1. Flow chart of this analysis based on CHARLS.**

**Independent variable.** The CHARLS database asks the following question: "Have you attended an educational or training course in the past month?" A "yes" answer is considered a continuing learner and is assigned a value of "1", otherwise it is "0".

**Covariate.** In order to control for the effects of confounding factors, 10 covariates were also chosen for this paper: Age, Gender, Residence, Marital status, Educational background [12], Self-assessed health [13], Cognitive [14], Sleep [15], Exercise [16] and Life satisfaction [17].

Dependent, independent and covariate definitions and attributes are shown in Table 1.

## Statistics analysis

All data were statistically analysed using STATA 17.0. Ordinary Least Squares (OLS) was used to explore the relationship between continued learning and employment among middle-aged and older adults. However, regression can only explain correlation and not causation. In addition there may be selection bias when using regression. Therefore propensity score matching (PSM) was chosen to address this issue. PSM is based on the Rubin causal model and the Neyman-Rubin causal model [18] and is designed to add covariates to artificially match the intervention group to an appropriate control

**Table 1. Definitions and attributes of dependent, independent and covariate variables.**

| Variable | Definition | Mean (SD) or frequency (%) |
|---|---|---|
| Employment | Yes "1" | 68.22 |
| | No "0" | 31.78 |
| Continue learning | Yes "1" | 1.00 |
| | No "0" | 99.00 |
| Age | >=45 years | 61.91(9.36) |
| Gender | Male "1" | 47.47 |
| | Female "0" | 52.53 |
| Residence | Rural "1" | 63.27 |
| | Urban "0" | 36.73 |
| Marital status | Married "1" | 86.53 |
| | Unmarried "0" | 13.47 |
| Educational background | Below primary school "1"<br>secondary schools "2"<br>junior high school "3"<br>High school or above "4" | 42.01<br>26.35<br>20.60<br>11.04 |
| Self-assessed health | Very poor "1"<br>Poor "2"<br>Fair "3"<br>Good "4"<br>Very good "5" | 5.70<br>18.87<br>51.36<br>12.39<br>11.69 |
| Cognitive* | 0 to 21 points, the higher the score the better the cognitive | 12.32(3.03) |
| Sleep | Time (hours) | 6.18(1.99) |
| Exercise | Yes "1" | 75.49 |
| | No "0" | 24.51 |
| Life satisfaction | Extremely satisfied "1"<br>Very satisfied "2"<br>Quite satisfied "3"<br>Not very satisfied "4"<br>Very dissatisfied "5" | 1.10<br>3.30<br>22.64<br>13.95<br>2.37 |

*Cognition was measured using the scale of Harmonised Cognitive Assessment Protocol.

group. Matching allows the sample distribution of the intervention and control groups to be close to a random distribution, thus reducing the effect of selection bias.

$$P(x_1, x_2 \ldots x_n) = Pr(D = 1|x_1, x_2 \ldots x_n) \tag{1}$$

$$Y_{1,i}, Y_{0,i}|x_1, x_2 \ldots x_n \tag{2}$$

Equation (1) explains the formula used to calculate the propensity score. The propensity score P $(x_1, x_2\ldots xn)$ is the probability that an individual will choose to receive learning and training (D = 1), conditional on the known factors x1, x2...xn. Equation (2) explains the principle of propensity score matching. After controlling for the factors $(x_1, x_2\ldots xn)$, the assignment of the intervention behaviour (D) can be considered random. We can then construct the propensity score by assigning behaviours (D) to these factors $(x_1, x_2\ldots xn)$. And after controlling for the propensity score, the assignment of intervention behaviours (D) can also be considered random. Based on the propensity score, we used three methods for matching to ensure robustness, including nearest neighbour matching with calipers, radius matching and kernel matching.

$$ATE = E(Y_{1i} - Y_{0i})$$

$$ATT = E(y \,|\, t = 1) - E(\_y \,|\, t = 1)$$

$$ATU = E(y \,|\, t = 0) - E(y \,|\, t = 0) \tag{3}$$

Finally, we measured treatment effects according to equation (3): the average treatment effect (ATE), the average treatment efficacy in the intervention group (ATT), and the treatment effect in the control (ATU), but focused mainly on the ATE.

## Results

### Analysis of independent and dependent variables based on age

In China, the proportion of people aged 45–59 who work is 82.1%, while the proportion of those who work after the age of 60 declines markedly, with 63.0% of those aged 60–74 working and only 32.9% of those aged 75 and over working. The corresponding proportion of the population continuing to study is very low, for example, only 1.8% of those aged 45–59 and 0.7% of those aged 60 and over. (See Table 2)

### OLS result

In order to verify the effect of continued learning of middle-aged and elderly people on employment, OSL was carried out, as well as OSL after the addition of covariates, as shown in Table 3.The results indicate that continued learning of middle-aged and elderly people significantly affects employment before and after the addition of the covariates (P < 0.001). However, this result cannot discharge selection bias.

### PSM and balance test

To rule out selection bias, we used logistic regression to calculate propensity scores and then matched appropriate subjects to the intervention group based on the propensity scores. As there are multiple matching models for PSM, this study focused on three dominant matching models, namely nearest neighbour matching, radius matching and kernel matching, to explore the effects of learning and participation in training on the employment of middle-aged and older adults.

In order to ensure the quality of matching and the reliability of the estimates, it was necessary to test the balancing assumption. According to the kernel density plot in Fig 2, it was demonstrated that the PSM effectively balanced the distribution of covariates between the intervention and control groups after PSM. To further analyse the matching, the mean deviation values were also calculated. Theoretically, the mean deviation after matching should be less than 10%. Table 4 reports the results of the balancing test for matching of the characteristic variables between the intervention and control groups, using nearest neighbour matching (k = 1) as an example. As can be seen in Table 3, the mean deviations of all covariates decreased significantly after matching and were contained within 10%.The t-test results indicate that no significant differences were observed between the two sample groups after matching, suggesting that the matching was effective, the quality of the matching was high, and the assumption of conditional independence was satisfied.

**Treatment effect**

Four matching modes were used in this study: nearest neighbour matching (k = 1, k = 4), radius matching and kernel matching. Nearest neighbour matching selects the nearest individual as the result of the match. When k = 1, it is a 1-to-1 nearest-neighbour match, and when k = 4, it is a 1-to-4 nearest-neighbour match. "Radius Matching" sets a radius value (0.01) and samples that fall within that radius are defined as similar samples. Caliper matching pre-sets the caliper range, and when matching is performed, only the control group individuals closest to the intervention group individuals are matched within the specified caliper range.

The results of average treatment effect on the treated (ATT) are shown in Table 5, where the ATT values show a positive impact of learning and training on the employment of middle-aged and older people in all the matching modes. The nearest neighbour match (K = 1) was significant at the 5% level and the other matches were significant at the 1% level.

To improve the stability of the estimation, multiple PSM models are generated by Bootstrap, which results in multiple propensity scores and matching results. In turn, the stability of the propensity score and matching effect is estimated.

Table 2. Age analysis of independent and dependent variables.

| Age (N = 43357) | Work (N = 29578) | Continuing learning (N = 435) |
|---|---|---|
| 45-59(N = 18809) | 82.1% (N = 15459) | 1.8% (N = 333) |
| 60-74(N = 20070) | 63.0% (N = 12646) | 0.5% (N = 95) |
| >75 (N = 4478) | 32.9% (N = 1473) | 0.2% (N = 7) |

Table 3. Results of OSL.

| Variable | Employment | | | Employment(Adding covariates) | | |
|---|---|---|---|---|---|---|
| | Coefficient | Std. err. | P-value | Coefficient | Std. err. | P-value |
| Continue learning | 0.168 | 0.017 | <0.001 | 0.084 | 0.020 | <0.001 |
| Age | | | | -0.017 | 0.001 | <0.001 |
| Gender | | | | 0.153 | 0.004 | <0.001 |
| Residence | | | | 0.199 | 0.004 | <0.001 |
| Marital status | | | | 0.082 | 0.007 | <0.001 |
| Educational background | | | | -0.039 | 0.002 | <0.001 |
| Self-assessed health | | | | 0.045 | 0.002 | <0.001 |
| Cognitive | | | | -0.005 | 0.001 | <0.001 |
| Sleep | | | | -0.001 | 0.001 | 0.508 |
| Exercise | | | | 0.081 | 0.005 | <0.001 |
| Life satisfaction | | | | -0.035 | 0.007 | 0.008 |

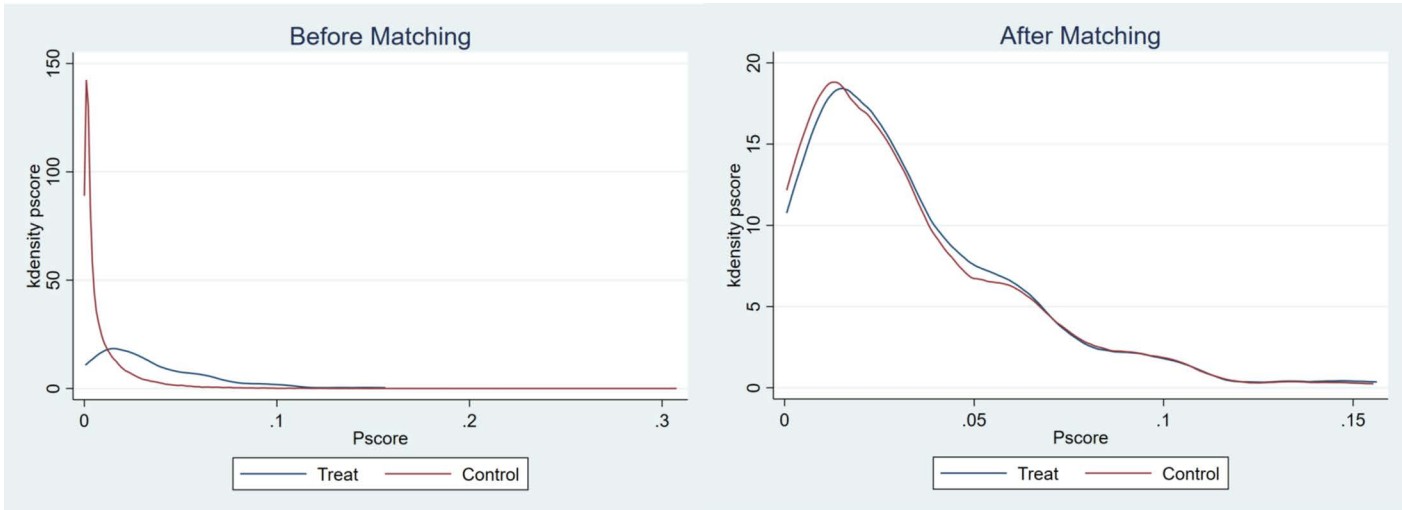

**Fig 2. Kernel density plots before and after matching.**

**Table 4. Balance test results.**

| Variables | U/M | %bias | P-value |
|---|---|---|---|
| Age | U | -76.4 | <0.001*** |
| | M | -2.0 | 0.728 |
| Gender | U | 40.9 | <0.001*** |
| | M | -1.4 | 0.828 |
| Residence | U | -33.2 | <0.001*** |
| | M | -2.3 | 0.753 |
| Marital status | U | 16.9 | 0.001*** |
| | M | -2.2 | 0.707 |
| Educational background | U | 97.4 | <0.001*** |
| | M | -3.8 | 0.560 |
| Self-assessed health | U | 82.7 | <0.001*** |
| | M | 2.4 | 0.697 |
| Cognitive | U | 72.8 | <0.001*** |
| | M | 4.2 | 0.666 |
| Sleep | U | 12.5 | 0.020** |
| | M | 7.7 | 0.210 |
| Exercise | U | 16.9 | 0.001*** |
| | M | 2.3 | 0.725 |
| Life satisfaction | U | 4.6 | 0.342 |
| | M | -4.7 | 0.473 |

Note:*U* unmatched, *M* matched

***p < 0.01,

**p < 0.05

**Table 5. The result of ATT.**

| Matching method | ATT | | |
|---|---|---|---|
| | Treated | Control | T-value |
| nearest neighbour matching(K = 1) | 0.848 | 0.793 | 2.1** |
| nearest neighbour matching(K = 4) | 0.848 | 0.792 | 2.7*** |
| radius matching(0.01) | 0.848 | 0.773 | 4.2*** |
| kernel matching | 0.848 | 0.725 | 7.0*** |

Note:

*1.67–1.96;

**1.96–2.56;

***>2.56

**Table 6. Results of the Bootstrap test.**

| Matching method | Treatment effect | Coefficient | Std. err. | Z-value |
|---|---|---|---|---|
| nearest neighbour matching(K = 1) | ATT | 0.055 | 0.031 | 1.76* |
| | ATU | 0.087 | 0.041 | 2.12** |
| | ATE | 0.086 | 0.041 | 2.13** |
| nearest neighbour matching(K = 4) | ATT | 0.056 | 0.025 | 2.25** |
| | ATU | 0.090 | 0.039 | 2.29** |
| | ATE | 0.090 | 0.039 | 2.29** |

Note:

*1.67–1.96;

**1.96–2.56;

***>2.56

Table 6 shows the Bootstrap test results for nearest neighbour matching (repeated with put-back random sampling 500 times from the original sample), where the values of ATE and ATU show only minor differences from ATT, and both are statistically significant, indicating relatively robust results.

To assess the robustness of our findings, we conducted sensitivity analyses using alternative causal inference approaches: inverse probability weighting (IPW) and doubly robust estimation. These methods adjust for confounding through different mechanisms, allowing us to test whether the results are consistent across methodologies.

The results were consistent across methods: the estimated average treatment effect (ATE) was 0.086 (95% CI: 0.004–0.168) with PSM, 0.110 (95% CI: 0.025–0.195) with IPW, and 0.092 (95% CI: 0.019–0.165) with doubly robust estimation. The similarity of these estimates supports the robustness of our findings (See Table 7).

**Table 7. ATE estimates across PSM, IPW, and Doubly Robust methods.**

| Method | ATE | 95%CI | P |
|---|---|---|---|
| PSM | 0.086 | 0.004-0.168 | 0.039** |
| IPW | 0.110 | 0.025-0.195 | 0.011** |
| Doubly Robust | 0.092 | 0.019-0.165 | 0.013** |

Note:

**p < 0.05

## Discussion

Across the globe, the employment landscape for middle-aged and elderly individuals is undergoing notable shifts [19]. With advancements in healthcare and improved living standards, populations are aging rapidly, leading to a larger proportion of older adults in the workforce [20]. However, challenges such as age discrimination, changing job requirements, and evolving labor market dynamics pose barriers to sustained employment for this demographic cohort [21]. As the proportion of older adults in the population increases, there is a heightened urgency to ensure their continued engagement in the labor market [22]. This entails not only addressing age-related stereotypes and discriminatory practices but also adapting workplaces and employment policies to accommodate the evolving needs and capabilities of older workers [20]. Moreover, there is a growing emphasis on promoting lifelong learning and skill development among older adults to enhance their employability and adaptability in an ever-changing job market [23]. However, despite the imperative to promote continuing education initiatives, empirical evidence regarding its impact on the employment prospects of middle-aged and elderly individuals remains scarce.

The findings of this study shed light on the relationship between continuing learning and employment among individuals aged 45 and above. While the observed employment rate of 68.22% among this demographic in China underscores their significant presence in the workforce, the low participation rate of only 1% in continuing education programs raises questions about the potential link between ongoing learning and employment outcomes. However, employing a propensity score matching (PSM) analysis, this study reveals a significant positive correlation between continuing learning and the employment of middle-aged and elderly individuals. Through robustness analysis, these findings are further validated, suggesting that those who actively pursue continuing education or skills development are more likely to be employed, highlighting the potential benefits of lifelong learning programs in promoting labor force participation among the older population.

The findings of this study demonstrate a significant improvement in employment probability attributable to the intervention, with estimates ranging from 5.5 to 12.3 percentage points across different matching methods. Specifically, the treatment group exhibited a mean employment probability of 0.848, compared to 0.725–0.793 in the control group, depending on the matching approach. Notably, kernel matching revealed the largest absolute improvement (0.848 vs. 0.725, $\Delta = 0.123$), representing a 17% relative increase in employment probability (0.123/0.725). Even the most conservative estimate, derived from nearest-neighbor matching ($\Delta = 0.055$), underscores the intervention's practical relevance and potential for societal impact if implemented at scale.These results are particularly compelling when contextualized within the broader literature on labor market interventions. For example, McKenzie's [24] review finds that the employment impact of vocational training is about 2–3 out of every 100 people who find a job. Agarwal and Mani's [25] meta-analysis reveals that the average employment impact of vocational training and apprenticeship programmes is 4 percentage points (with confidence intervals ranging from 2 to 6 percentage points). This suggests that the intervention under study is a robust and viable policy option for enhancing employment outcomes.

Continuing learning has a positive impact on employment, driven by multiple mechanisms. Firstly, it likely enhances technical skills, equipping individuals with the up-to-date knowledge required in the labor market. As Rodriguez et al. [26] mentioned, continuing education provides individuals with the latest technical knowledge needed in the labor market. Secondly, it may foster soft skills, such as communication and adaptability, which are increasingly valued by employers. Jin & Baumgartner [27] pointed out that soft skills like communication and adaptability are highly valued by employers in today's workplace. Thirdly, participation in continuing learning may signal commitment to professional development. Desjardins et al. [28] suggested that participating in continuing learning can signal one's commitment to professional development to employers.

Despite these benefits, the participation rate in continuing learning remains low, at just 1%. This can be attributed to barriers such as financial constraints and time limitations, particularly due to family responsibilities. Kogovšek et al. [29] found that financial constraints and time limitations, particularly due to family responsibilities, are significant barriers to

continuing learning. Additionally, a lack of awareness about digital learning resources further restricts access, especially for rural populations. Gates & Wilson-Menzfeld [30] noted that rural populations often lack awareness of available digital learning resources, further limiting their access to continuing learning opportunities. Davies et al. [31] noted that digitization and advanced data analytics have created a skills gap that further impacts employment.

To address these issues, policymakers should take several measures. Firstly, expand access to programs in rural areas to ensure that remote workers can also obtain necessary learning resources. Dello Russo et al. [32] recommended expanding access to learning programs in rural areas to enhance the reach of continuing learning. Secondly, collaborate with employers to design age-inclusive training programs that meet the learning needs of employees at different stages of their careers. By collaborating with employers, training programs can be tailored to meet the specific needs of different age groups. Thirdly, raise public awareness of digital learning resources through campaigns and education to eliminate information barriers. Increasing awareness of digital learning resources can help overcome some of the barriers faced by rural populations and those with limited access to traditional learning opportunities. Finally, by implementing advanced causal learning techniques [33], policymakers can maximize the impact of lifelong learning on employment outcomes and enhance the adaptability and competitiveness of the workforce.

## Conclusion

In conclusion, the findings of this study underscore the importance of continuing learning in facilitating the employment prospects of middle-aged and elderly individuals. As populations continue to age and workforce dynamics evolve, investing in lifelong learning initiatives becomes imperative in ensuring the continued engagement and productivity of older workers in the labor market.By addressing barriers to educational participation and promoting lifelong learning, policymakers, educators, and employers can empower older populations to remain active contributors to the workforce and society at large.

## Acknowledgments

We express our gratitude to the CHARLS team for providing us with the data, and we extend our appreciation to every respondent in the study for their valuable contributions.

## Author contributions

**Conceptualization:** Yuan TAO.

**Data curation:** Yuan TAO.

**Funding acquisition:** Kexin Ren.

**Methodology:** Kexin Ren.

**Writing – original draft:** Yuan TAO.

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
