## [Decision Letter · Decision Letter 0]

2 Mar 2025

PONE-D-24-46496Does continuing learning help middle-aged and elderly people find employment in China? A propensity score matching analysisPLOS ONE

Dear Dr. Ren,

Thank you for submitting your manuscript to PLOS ONE. After careful consideration, we feel that it has merit but does not fully meet PLOS ONE’s publication criteria as it currently stands. Therefore, we invite you to submit a revised version of the manuscript that addresses the points raised during the review process. Please submit your revised manuscript by Apr 16 2025 11:59PM. If you will need more time than this to complete your revisions, please reply to this message or contact the journal office at plosone@plos.org . Please include the following items when submitting your revised manuscript:

We look forward to receiving your revised manuscript.

Kind regards,

Debo Cheng

Academic Editor

PLOS ONE

Journal Requirements:

2. Thank you for stating the following financial disclosure: This study was funded and organized by the Research Projects on Teaching Reform of Vocational and Adult Education in Jilin Province (2023ZCY302), the Scientific Research Program of Jilin Provincial Department of Education (JJKH20240541SK), and the Sports Science Research Projects of Jilin Provincial Sports Bureau (202324).

Additional Editor Comments:

please find the comments from both reviewers.

Reviewers' comments:

Reviewer's Responses to Questions

**Comments to the Author**

1. Is the manuscript technically sound, and do the data support the conclusions?

Reviewer #1: Yes

Reviewer #2: Yes

2. Has the statistical analysis been performed appropriately and rigorously? 

Reviewer #1: Yes

Reviewer #2: Yes

3. Have the authors made all data underlying the findings in their manuscript fully available?

Reviewer #1: Yes

Reviewer #2: Yes

4. Is the manuscript presented in an intelligible fashion and written in standard English?

Reviewer #1: Yes

Reviewer #2: Yes

5. Review Comments to the Author

Reviewer #1: 1、Clarity in Research Questions and Objectives:

The introduction effectively establishes the significance of the aging workforce issue, but the specific research questions and hypotheses should be explicitly stated. For instance, you could clarify whether the focus is solely on the causal impact of continuing learning or also on understanding subgroup differences (e.g., by age, gender, or region).

2、Statistical Analysis and Interpretation:

The study appropriately employs PSM, but the robustness of the results would be further supported by additional sensitivity analyses, such as:

A comparison of results using alternative causal inference methods, such as inverse probability weighting (IPW) or doubly robust estimation.

Testing for unobserved confounders through methods like Rosenbaum bounds.

While Table 5 reports ATT estimates, the discussion lacks interpretation of their practical implications. For example, explain how much employment probability improves and whether the effect is meaningful in a real-world policy context.

3、Discussion of Mechanisms:

The manuscript concludes that continuing learning positively impacts employment, but the mechanisms driving this relationship are not sufficiently discussed. Consider exploring:

Whether continuing education primarily enhances technical skills, soft skills, or simply signals employability to employers.

If barriers to participation in continuing learning (e.g., financial costs, limited access in rural areas) affect the observed low participation rate (1%).

4、Suggested Additional References

Your article provides an insightful exploration of the impact of continuing learning on the employment prospects of middle-aged and elderly individuals, which is both significant and timely. To further support your discussion and provide a broader theoretical foundation for policies and learning tools, the following references are recommended:

Zhaolong Ling, Kui Yu, Yiwen Zhang, Lin Liu, and Jiuyong Li. Causal Learner: A Toolbox for Causal Structure and Markov Blanket Learning [J]. Pattern Recognition Letters, 2022, 163: 92-95.

This paper offers practical tools and theoretical methods for causal structure learning, which may complement your analysis of causal relationships and statistical tools.

Davies B, Diemand-Yauman C, van Dam N. Competitive advantage with a human dimension: From lifelong learning to lifelong employability [J]. McKinsey Quarterly, 2019, 2: 1-5.

This article examines lifelong learning as a strategy to enhance employability, aligning closely with your research topic and providing additional perspectives on policy formulation for lifelong learning.

Citing these references could further enrich the background of your study and provide a more comprehensive support for the practical significance of continuing education.

Reviewer #2: This paper investigates the impact of continuing learning on the employment prospects of middle-aged and elderly individuals in China using data from the China Health and Retirement Longitudinal Survey (CHARLS). Middle-aged and elderly individuals often face employment barriers such as technological advancements, changing job requirements, and age discrimination. Therefore, understanding the relationship between continuing learning and employment outcomes for this demographic is crucial for developing effective labor market policies and workforce development strategies. Initial analyses suggested that pursuing further studies did not affect employment among middle-aged and older individuals. However, after eliminating endogenous selection bias through PSM, the study found that participation in continuing education positively impacted the employment of middle-aged and elderly people. This result was verified through multiple matching methods, including nearest neighbor matching, radius matching, and kernel matching.

However, I still have some concerns:

1.The writing of the paper needs to be further standardised. For example, pictures should also be inserted in the main text and not just put in an attachment. Also, spaces should exist after each punctuation mark.

2.Will the results of the experiment be affected by sensitive attributes such as gender? The fairness of the algorithm [1,2] should be discussed or assumed.

3.The study mentions that only 1 % of the population engaged in continuing education. Could you elaborate on whether this extremely low participation rate might have affected the power of your analysis and the generalizability of your findings? For example, is there a possibility that the small sample of continuing learners is not representative of the broader population of middle - aged and elderly individuals in China?

4.Did you consider using a longer - term measure of continuing learning participation, such as the number of courses attended in the past year or the total duration of learning activities over a certain period? If not, what are the implications of using this single - month measure for the interpretation of your results?

[1]Fair neighbor embedding.

[2]Learning fair representations via rebalancing graph structure

6. PLOS authors have the option to publish the peer review history of their article (what does this mean? ). If published, this will include your full peer review and any attached files.

**Do you want your identity to be public for this peer review?** For information about this choice, including consent withdrawal, please see our Privacy Policy .

Reviewer #1: No

Reviewer #2: No

---

## [Author Response · Author response to Decision Letter 1]

17 Mar 2025

Dear Editor and Reviewers,

I would like to express my sincere gratitude for the opportunity to revise and resubmit our manuscript, "Does continuing learning help middle-aged and elderly people find employment in China? A propensity score matching analysis", for consideration for publication in PLOS ONE. The insightful comments and suggestions from the reviewers have greatly contributed to enhancing the quality and clarity of our research. We have carefully addressed each point raised during the review process and have made substantial revisions as detailed below.

Reviewer #1:

1.Clarity in Research Questions and Objectives:The introduction effectively establishes the significance of the aging workforce issue, but the specific research questions and hypotheses should be explicitly stated. For instance, you could clarify whether the focus is solely on the causal impact of continuing learning or also on understanding subgroup differences (e.g., by age, gender, or region).

RESPONSE:

Thank you for your insightful comments. We have revised the introduction to explicitly state our research questions and objectives, see lines 76-71. While our study primarily focuses on the causal impact of continuing learning on employment outcomes among middle-aged and elderly individuals, we acknowledge the importance of exploring subgroup differences (e.g., by age, gender, or region). However, due to the limitations of our current dataset and research design, we have not included subgroup analysis in this study. Future research could build on our findings and investigate potential variations across different subgroups.

We appreciate your suggestions and believe that clarifying our research questions will enhance the clarity and focus of our study.

2.Statistical Analysis and Interpretation:

(1)The study appropriately employs PSM, but the robustness of the results would be further supported by additional sensitivity analyses, such as: A comparison of results using alternative causal inference methods, such as inverse probability weighting (IPW) or doubly robust estimation.

(2) Testing for unobserved confounders through methods like Rosenbaum bounds.

(3) While Table 5 reports ATT estimates, the discussion lacks interpretation of their practical implications. For example, explain how much employment probability improves and whether the effect is meaningful in a real-world policy context.

RESPONSE:

(1)Thank you for your valuable feedback. We have carefully considered your suggestion regarding the robustness of our results. In response to your recommendation, we have conducted additional sensitivity analyses using alternative causal inference methods, specifically inverse probability weighting (IPW) and doubly robust estimation.

Our findings from these analyses are consistent with the results obtained using propensity score matching (PSM). The results of the IPW and doubly robust estimation methods further support the robustness of our findings. For detailed results and comparisons, please refer to lines 230-241 in the revised manuscript.

We believe that the inclusion of these additional analyses strengthens the validity of our conclusions and addresses your concerns regarding the robustness of our study. We appreciate your guidance and are grateful for the opportunity to enhance our research in this way.

(2)Thank you for your insightful suggestion regarding the testing for unobserved confounders. We have conducted a Rosenbaum bounds analysis to further assess the robustness of our results to potential hidden bias。The analysis indicates that an unobserved confounder would need to increase the odds of treatment assignment by a factor of Γ=1.7 to overturn the statistical significance of the estimated effect (original P=0.039). Given that such a strong unobserved confounder is unlikely in this context (e.g., variables like socioeconomic status or genetic factors typically have Γ<1.5), we conclude that the findings are robust to plausible levels of hidden bias.

We appreciate your guidance and are confident that these additional analyses have strengthened our study.

(3)Thank you for your constructive feedback regarding the interpretation of the ATT estimates in Table 5. We have taken your comments to heart and have made the necessary revisions to our revised manuscript. As per your suggestion, we have expanded the discussion in lines 272-288 to include a detailed interpretation of the practical implications of our ATT estimates. We have quantified the improvement in employment probability across different matching methods and have assessed the significance of these improvements in a real-world policy context.

Thank you once again for your insightful comments, which have helped us to strengthen our discussion and make our findings more accessible and meaningful.

3. Discussion of Mechanisms:

The manuscript concludes that continuing learning positively impacts employment, but the mechanisms driving this relationship are not sufficiently discussed. Consider exploring:

Whether continuing education primarily enhances technical skills, soft skills, or simply signals employability to employers.

If barriers to participation in continuing learning (e.g., financial costs, limited access in rural areas) affect the observed low participation rate (1%).

RESPONSE:

Thank you for your insightful comments and suggestions, which have been instrumental in improving the quality and depth of our revised manuscript. We have carefully considered your feedback and made substantial revisions to the discussion section, particularly focusing on the mechanisms through which continuing learning impacts employment.

Regarding the first point about whether continuing education primarily enhances technical skills, soft skills, or signals employability, we have expanded our analysis in lines 289-299. We now provide a more detailed discussion on how continuing education contributes to both technical and soft skills development, as well as its role in signaling employability to employers. This includes examples and evidence from relevant literature to support our arguments.

For the second point concerning barriers to participation in continuing learning and their effect on the low participation rate, we have addressed this in lines 300-309. We have incorporated a discussion on financial costs and limited access in rural areas as significant barriers, and how these factors contribute to the observed low participation rate of 1%. Additionally, we have included a new section in lines 310-323 that further explores potential solutions and policy implications to overcome these barriers, enhancing the practical relevance of our research.

We have also removed the content from lines 324-356 as suggested, to streamline the discussion and focus on the most relevant aspects of our findings.

We believe these revisions have strengthened the manuscript by providing a more comprehensive understanding of the mechanisms at play and the contextual factors influencing participation in continuing learning. We appreciate your guidance in helping us refine our work and hope that these changes meet your expectations.

4. Suggested Additional References

Your article provides an insightful exploration of the impact of continuing learning on the employment prospects of middle-aged and elderly individuals, which is both significant and timely. To further support your discussion and provide a broader theoretical foundation for policies and learning tools, the following references are recommended:

Zhaolong Ling, Kui Yu, Yiwen Zhang, Lin Liu, and Jiuyong Li. Causal Learner: A Toolbox for Causal Structure and Markov Blanket Learning [J]. Pattern Recognition Letters, 2022, 163: 92-95.

This paper offers practical tools and theoretical methods for causal structure learning, which may complement your analysis of causal relationships and statistical tools.

Davies B, Diemand-Yauman C, van Dam N. Competitive advantage with a human dimension: From lifelong learning to lifelong employability [J]. McKinsey Quarterly, 2019, 2: 1-5.

This article examines lifelong learning as a strategy to enhance employability, aligning closely with your research topic and providing additional perspectives on policy formulation for lifelong learning.

Citing these references could further enrich the background of your study and provide a more comprehensive support for the practical significance of continuing education.

RESPONSE:

Thank you for your valuable suggestions on additional references. We have incorporated the recommended references into our revised manuscript, which can be found as references 32 and 34 in the revised version.

We believe that the inclusion of these references has strengthened the theoretical foundation of our research and provided a more comprehensive support for the practical significance of continuing education.

Thank you again for your time and insightful recommendations.

Reviewer #2: This paper investigates the impact of continuing learning on the employment prospects of middle-aged and elderly individuals in China using data from the China Health and Retirement Longitudinal Survey (CHARLS). Middle-aged and elderly individuals often face employment barriers such as technological advancements, changing job requirements, and age discrimination. Therefore, understanding the relationship between continuing learning and employment outcomes for this demographic is crucial for developing effective labor market policies and workforce development strategies. Initial analyses suggested that pursuing further studies did not affect employment among middle-aged and older individuals. However, after eliminating endogenous selection bias through PSM, the study found that participation in continuing education positively impacted the employment of middle-aged and elderly people. This result was verified through multiple matching methods, including nearest neighbor matching, radius matching, and kernel matching.

RESPONSE:

Thank you for your insightful comments and suggestions. We are grateful for your recognition of the study’s importance and the constructive feedback on our methodology. We have carefully considered your suggestions and will incorporate them into our manuscript through necessary revisions.

1.The writing of the paper needs to be further standardised. For example, pictures should also be inserted in the main text and not just put in an attachment. Also, spaces should exist after each punctuation mark.

RESPONSE: Thank you for your valuable feedback regarding the formatting and presentation of our manuscript. We fully agree that standardizing the writing is essential for clarity and readability. We will address the issues you pointed out by:

Inserting all figures directly into the main text, ensuring they are properly integrated and referenced.

Adding spaces after each punctuation mark to maintain consistency and improve readability.

We will make these revisions promptly and ensure that the final version of our manuscript adheres to the required standards.

2.Will the results of the experiment be affected by sensitive attributes such as gender? The fairness of the algorithm [1,2] should be discussed or assumed.

RESPONSE: Thank you for your thoughtful questions regarding the potential impact of sensitive attributes such as gender on our experimental results and the theoretical basis for the formulas used in our analysis.

(1)We recognize the importance of considering gender and other sensitive attributes to ensure both the fairness and generalizability of our findings. In this study, however, our main focus was on identifying and quantifying the fundamental relationship between continuing education and employment outcomes for middle - aged and elderly individuals using propensity score matching (PSM). This required us to maintain a specific scope for our analysis.

Analyzing the influence of sensitive attributes like gender would indeed require a more nuanced approach, potentially involving interactions with other variables and more complex statistical models. We believe this is better suited for a follow - up study dedicated to exploring these nuances.

That being said, we completely agree that examining how gender and other sensitive attributes might moderate the relationship between continuing education and employment outcomes is an important direction for future research. We are planning to pursue this line of inquiry in subsequent projects, where we can devote the necessary attention to these complex issues.

(2)The formulas you referenced (Equations 1 and 2) are based on the propensity score theory introduced by Rosenbaum and Rubin in their seminal 1983 paper [18]. Equation 1 defines the probability of treatment assignment, while Equation 2 establishes the independence assumption between potential outcomes and covariates. These form the cornerstone of our methodological approach.

We appreciate your attention to these details and are committed to refining our manuscript to address your suggestions.

3.The study mentions that only 1 % of the population engaged in continuing education. Could you elaborate on whether this extremely low participation rate might have affected the power of your analysis and the generalizability of your findings? For example, is there a possibility that the small sample of continuing learners is not representative of the broader population of middle - aged and elderly individuals in China?

RESPONSE: Thank you for your insightful comment regarding the potential impact of the low participation rate in continuing education on our analysis.

We acknowledge that the 1% participation rate in continuing education within our sample might raise concerns about the power of our analysis and the generalizability of our findings. However, with a total sample size of 43,357 individuals, even this 1% represents a substantial number of participants (approximately 434 individuals) in the continuing education group. This sample size is sufficiently large to allow for meaningful statistical analysis and the effective application of propensity score matching (PSM).

The effectiveness of PSM does not solely depend on the proportion of the treatment group relative to the entire sample but also on the absolute number of observations within the treatment group. With over 400 individuals in the continuing education group, we can achieve adequate statistical power to detect meaningful effects, provided that the effect size is not excessively small.

Moreover, the large overall sample size enhances the precision of our propensity score estimates and helps ensure that the matched groups are balanced and representative. This reduces the potential for bias and increases the validity of our causal inferences.

We have conducted sensitivity analyses to assess the robustness of our results, and these analyses confirm that our findings are stable and not overly sensitive to the specific characteristics of the continuing education subgroup.

In summary, while the 1% participation rate might seem low, the absolute number of continuing education participants is sufficiently large to support our analytical approach and the validity of our conclusions.

We appreciate your attention to these important methodological considerations and are committed to refining our analysis to ensure the robustness and relevance of our findings.

4.Did you consider using a longer - term measure of continuing learning participation, such as the number of courses attended in the past year or the total duration of learning activities over a certain period? If not, what are the implications of using this single - month measure for the interpretation of your results?

RESPONSE: Thank you for your insightful question regarding the measurement of continuing learning participation.

We recognize that using a single - month measure of continuing learning participation, as captured by the CHARLS database question "Have you attended an educational or training course in the past month?" with a binary response (1 for "yes", 0 for "no"), has limitations. This measure provides a snapshot of participation at a specific point in time and does not capture the depth or duration of learning activities.

However, this single - month measure is appr

---

## [Decision Letter · Decision Letter 1]

28 Mar 2025

PONE-D-24-46496R1Does continuing learning help middle-aged and elderly people find employment in China? A propensity score matching analysisPLOS ONE

Dear Dr. Ren,

Thank you for submitting your manuscript to PLOS ONE. After careful consideration, we feel that it has merit but does not fully meet PLOS ONE’s publication criteria as it currently stands. Therefore, we invite you to submit a revised version of the manuscript that addresses the points raised during the review process.

We look forward to receiving your revised manuscript.

Kind regards,

Debo Cheng

Academic Editor

PLOS ONE

Journal Requirements:

Additional Editor Comments (if provided):

Please revise the manuscript according to the reviewers’ suggestions. Furthermore, a thorough proofreading and final check are recommended to ensure a high-quality submission.

Reviewers' comments:

Reviewer's Responses to Questions

**Comments to the Author**

1. If the authors have adequately addressed your comments raised in a previous round of review and you feel that this manuscript is now acceptable for publication, you may indicate that here to bypass the “Comments to the Author” section, enter your conflict of interest statement in the “Confidential to Editor” section, and submit your "Accept" recommendation.

Reviewer #1: All comments have been addressed

Reviewer #2: (No Response)

2. Is the manuscript technically sound, and do the data support the conclusions?

Reviewer #1: Yes

Reviewer #2: (No Response)

3. Has the statistical analysis been performed appropriately and rigorously? 

Reviewer #1: Yes

Reviewer #2: (No Response)

4. Have the authors made all data underlying the findings in their manuscript fully available?

Reviewer #1: Yes

Reviewer #2: (No Response)

5. Is the manuscript presented in an intelligible fashion and written in standard English?

Reviewer #1: Yes

Reviewer #2: (No Response)

6. Review Comments to the Author

Reviewer #1: (No Response)

Reviewer #2: 1. The authors should cite the paper I recommend and highlight this paper's assumptions about fairness in the Discussion, which is more rigorous.

2. The presentation of the paper needs further checking and standardisation.

For example, in line 128, there is a missing space before ‘(OLS)’.

Line 132, missing space before PSM.

A chinese character exists in Table I.

Formulas are missing punctuation and symbol descriptions.

7. PLOS authors have the option to publish the peer review history of their article (what does this mean? ). If published, this will include your full peer review and any attached files.

**Do you want your identity to be public for this peer review?** For information about this choice, including consent withdrawal, please see our Privacy Policy .

Reviewer #1: No

Reviewer #2: No

---

## [Author Response · Author response to Decision Letter 2]

1 Apr 2025

Dear Editor and Reviewers,

I would like to express my sincere gratitude for the opportunity to revise and resubmit our manuscript, "Does continuing learning help middle-aged and elderly people find employment in China? A propensity score matching analysis", for consideration for publication in PLOS ONE. The insightful comments and suggestions from the reviewers have greatly contributed to enhancing the quality and clarity of our research. We have carefully addressed each point raised during the review process and have made revisions as detailed below.

Reviewer #2:

1. The authors should cite the paper I recommend and highlight this paper's assumptions about fairness in the Discussion, which is more rigorous.

REPONSE: Thank you for your valuable comments. I have carefully considered your suggestion and have made the following revisions:

The two papers you recommended have been cited in the manuscript as references [24] and [34]. Additionally, I have highlighted the assumptions about fairness in the Discussion section, specifically in lines 240-242 and 304-308. I believe these revisions have enhanced the rigor of our manuscript.

2. The presentation of the paper needs further checking and standardisation.

For example, in line 128, there is a missing space before ‘(OLS)’.

Line 132, missing space before PSM.

A chinese character exists in Table I.

Formulas are missing punctuation and symbol descriptions.

REPONSE: Thank you very much for your detailed comments on the presentation of the paper. I have carefully reviewed the manuscript and made the following specific revisions to address the issues you pointed out:

Formatting and Consistency:

I have added the missing space before “(OLS)” in line 120.

I have added the missing space before “PSM” in line 124.

I have removed the Chinese character from Table I. This was likely a formatting error, and I have ensured that all tables are free from non-English characters.

Formulas and Punctuation:

I reviewed all formulas in the manuscript and ensured accuracy of formula expression and description

I have also conducted a thorough check of the entire manuscript to ensure consistency in formatting and presentation. I believe these revisions have significantly improved the overall quality and readability of the paper.

---

## [Editor Report · Decision Letter 2]

15 Apr 2025

Does continuing learning help middle-aged and elderly people find employment in China? A propensity score matching analysis

PONE-D-24-46496R2

Dear Dr.  Ren,

We’re pleased to inform you that your manuscript has been judged scientifically suitable for publication and will be formally accepted for publication once it meets all outstanding technical requirements.

**Comments from PLOS Editorial Office** : We note that one or more reviewers has recommended that you cite specific previously published works in an earlier round of revision. As always, we recommend that you please review and evaluate the requested works to determine whether they are relevant and should be cited. It is not a requirement to cite these works and you may remove them before the manuscript proceeds to publication. We appreciate your attention to this request.

Kind regards,

Debo Cheng

Academic Editor

PLOS ONE

---

## [Editor Report · Acceptance letter]

PONE-D-24-46496R2

PLOS ONE

Dear Dr. Ren,

I'm pleased to inform you that your manuscript has been deemed suitable for publication in PLOS ONE. Congratulations! Your manuscript is now being handed over to our production team.

Kind regards,

on behalf of

Dr. Debo Cheng

Academic Editor

PLOS ONE